# Potential Applications of Chitosan-Based Nanomaterials to Surpass the Gastrointestinal Physiological Obstacles and Enhance the Intestinal Drug Absorption

**DOI:** 10.3390/pharmaceutics13060887

**Published:** 2021-06-15

**Authors:** Nutthapoom Pathomthongtaweechai, Chatchai Muanprasat

**Affiliations:** Chakri Naruebodindra Medical Institute, Faculty of Medicine Ramathibodi Hospital, Mahidol University, Bang Phli 10540, Samut Prakan, Thailand; chatchai.mua@mahidol.ac.th

**Keywords:** chitosan, drug delivery, drug absorption, intestinal assimilation, oral bioavailability

## Abstract

The small intestine provides the major site for the absorption of numerous orally administered drugs. However, before reaching to the systemic circulation to exert beneficial pharmacological activities, the oral drug delivery is hindered by poor absorption/metabolic instability of the drugs in gastrointestinal (GI) tract and the presence of the mucus layer overlying intestinal epithelium. Therefore, a polymeric drug delivery system has emerged as a robust approach to enhance oral drug bioavailability and intestinal drug absorption. Chitosan, a cationic polymer derived from chitin, and its derivatives have received remarkable attention to serve as a promising drug carrier, chiefly owing to their versatile, biocompatible, biodegradable, and non-toxic properties. Several types of chitosan-based drug delivery systems have been developed, including chemical modification, conjugates, capsules, and hybrids. They have been shown to be effective in improving intestinal assimilation of several types of drugs, e.g., antidiabetic, anticancer, antimicrobial, and anti-inflammatory drugs. In this review, the physiological challenges affecting intestinal drug absorption and the effects of chitosan on those parameters impacting on oral bioavailability are summarized. More appreciably, types of chitosan-based nanomaterials enhancing intestinal drug absorption and their mechanisms, as well as potential applications in diabetes, cancers, infections, and inflammation, are highlighted. The future perspective of chitosan applications is also discussed.

## 1. Introduction

Oral entry route is the most favorable and frequently used pathway for drug administration owing to its simplicity, feasibility, convenience, low-cost manufacturing process, non-invasiveness, and safety for most patients [1,2]. After intake of medication, the orally administered drugs travel down to the stomach and they are majorly absorbed at the proximal part of small intestine before entering the systemic circulation. However, the drug substances have to challenge with the harsh environment of the gastrointestinal (GI) tract, including gastric pH, digestive enzymes, bile salts, GI motility, and intestinal mucosal layers, ameliorating the drug absorption or drug assimilation and leading to poor oral bioavailability [3,4]. In order to overcome these physiological obstacles, the drug delivery system has been designed and developed. The ideal characteristics of nano-drug carriers are safe, biocompatible, biodegradable, acid-tolerant, and GI enzyme-stable [5,6,7]. Moreover, to deliver the drugs to the systemic circulation, the carriers should help to penetrate the mucosal microenvironment and prolong the drug-mucosa contact time for extent duration of drug absorption [8,9]. In response to the fluctuation of pH along the GI tract, the suitable carriers should be able to control the drug release in each GI segment. In recent times, the investigators have taken the advantages of these properties found in the natural polymers, particularly chitosan [2,4].

Chitosan is a cationic linear heteropolysaccharide, consisting of β-(1,4)-linked D-glucosamine (GlcN, deacetylated monomer) and N-acetyl-D-glucosamine (GlcNAc, acetylated monomer) [10,11]. It is obtained from chitin, containing β-(1,4)-linked 2-acetamido-2-deoxy-β-D-glucose, which is the second-most abundant natural polymer in the world and is mostly found in the structural component of exoskeletons of crustaceans (e.g., shrimps, lobsters, and crabs), insects, and arthropods as well as in the cell wall of fungi [10,11]. The outstanding features of chitosan, including natural abundance, water solubility, non-toxicity, biodegradability, and chemical modifiability, make this natural biomaterial attractive as an ideal candidate for adopting in pharmacological and biotechnological applications [8]. In the drug delivery system, the chitosan-based nanoparticles have been formulated by several techniques, such as chemical modification, ionic gelation, and polyelectrolyte complexation methods [12,13]. These nanomaterials can boost up the oral drug absorption by improving mucoadhesion, the permeation-enhancing effect, and controlled drug release [2,4,8,9].

Herein, the aims of this review are (i) to provide the comprehensive knowledge involving GI physiological challenge-modulated intestinal drug absorption and oral bioavailability, as well as (ii), to address the implication of chitosan and various types of emerged chitosan-based nanomaterials in the improvement of intestinal drug assimilation.

## 2. The GI Challenges of Intestinal Drug Absorption

In any given patients, the orally taken drugs must encounter a ruthless GI environment, which is influenced by multiple determinants, such as the age, gender, and ethnicity of the treated patients [14,15,16,17]. These factors can affect the physiologically challenging variables involved in the drug absorption, including intraluminal pH, gastric emptying time, intestinal transit time, GI motility, intestinal transporter proteins, gut microbiota, and disease conditions (Figure 1).

Age-dependent alteration of oral drug absorption gives rise to the adjustment of drug dosing to meet the medical requirement, particularly in childhood, due to the immaturity of the GI tract [14,18]. Therefore, tailor-made pediatric dosage at a specific age is necessary. In addition, poor cooperation in pediatric patients makes it more difficult. For the elderly, the frailty is associated with the progressive impairment of organ structures and functions, which affects the GI physiology and oral drug bioavailability [15,19]. Additionally, polypharmacy in advancing age brings about the increased risk of adverse drug reactions and poor drug compliance [19]. Gender-based changes in GI function also needs to be considered. The difference in occupational exposures, behavior, lifestyle, and medications between men and women leads to the dissimilarity in body weight, body surface area, and the amount of water, contributing to sex-divergent pharmacokinetics [20,21]. Ethnicity/race-related differences in body response to medications has been documented due to the interaction of genetic variation and the environmental factors [17,22]. In particular, the racial genetic difference is resulted from the variance in genetic polymorphisms [23,24].

The GI physiological challenges limit the capability of absorption of orally taken drugs and thus affect their oral bioavailability. The amount of drugs that are assimilated at the absorption site will reflect the alteration of the pharmacokinetics in the patients, compared with normal subjects. A synopsis of the physiological challenges impacting on oral drug bioavailability is recapitulated as below and is depicted in Figure 1. In addition, the effect of chitosan on those parameters and disease states are included.

### 2.1. Gastric pH

The gastric pH is acidic, with a pH between 1.5 and 3.5. When passing down to the small intestine, the pH slowly increases to 5–6 in the duodenum and 7–8 in the jejunum and terminal ileum. Meanwhile, the pH falls to 5.7–6.4 in the caecum and rises again at the range of 6.1–7.5 in the descending colon and rectum [14,25]. Therefore, the drugs have to expose to the variation of the pH along the GI tract, which may cause the deactivation of the drugs, particularly in protein and peptide drugs by the modulation of differential oxidation, hydrolysis, or deamination of these drugs [26]. In the drug delivery system, the environmental pH determines the drug dissolution and drug absorption by affecting the degree of ionization (pK_a_) of the drug substrates [27]. The water-insoluble weakly alkaline drugs are ionized and dissolved in the low-pH stomach lumen, but have poor drug absorption at the small intestine [28]. Impaired drug absorption of weak basic drug can occur in the stomach of the patients with achlorhydria or hypochlorhydria, which have no or decreased gastric acid secretion, respectively [29]. In addition, the patients who took weakly basic drugs with antacids, which could cause a rise in gastric pH, were reported to reduce the drug absorption of those basic drugs by the chelation with polyvalent cations (e.g., Ca^2+^, Mg^2+^, and Al^3+^) and subsequent formation of insoluble complex [30,31].

At birth, the gastric pH is neutral due to the presence of amniotic fluid in the stomach. It then gradually decreases until the age of two years, reaching to the adult-equivalent pH [32,33]. In the elderly, the gastric pH is increased, resulting in the slight decrease in drug absorption [19,33]. In women, smaller gastric secretion is found with pH ~ 0.5 units higher than in men. Therefore, the drugs that require the acidic gastric pH have a poorer bioavailability in women [20,34,35].

Chitosan has the amino group with the pK_a_ of ~6.5, relating to the degree of N-deacetylation, and is fully protonated at the pH of ~4, resulting in the increase in acidity of chitosan [36,37]. Therefore, the chitosan-loaded drug has better mucoadhesive and permeation-enhancing properties, promoting drug absorption at proximal part of GI tract, including stomach and duodenum, when compared with the chitosan-free group [4,8,37,38]. The problem is that chitosan precipitates at the pH of >6.5 in the jejunum, ileum, and colon, and thus chitosan has less adhesion to the mucus layer of GI tract, leading to the ineffective drug absorption [38,39]. Interestingly, this drawback is solved by the modification of chitosan, such as thiolated chitosan, which will be addressed in a later section [40,41,42].

### 2.2. GI Motility

The GI motility is defined as the contraction and relaxation of GI smooth muscles to propel the contents along the GI tract with the change in intraluminal pressure and it is a causal determinant for the drug absorption. The gastric emptying, which is a process of removing the content from the stomach and moving it into the duodenum, also has an impact on the intestinal assimilation [43,44]. The increasing rate of gastric emptying, found in the case of solutions or suspensions, may lead to a rise in the drug absorption rate [25,45,46]. However, medications for gastric ulcer need delayed gastric emptying in order to prolong in the stomach. After reaching the small intestine, the content of drugs in the bowel needs a sufficient residence time and a long intestinal transit time in order to facilitate the opportunity of drug absorption at this site, owing to the enormous surface area of the small intestine [47,48,49]. Peristaltic contraction also enhances the capacity of drug absorption, as it promotes drug dissolution and membrane–drug contact [50]. In addition, the volume of GI luminal fluids may contribute to drug assimilation by affecting drug dissolution and providing the driving force for drug permeation [51,52]. In adults, the rate of gastric emptying is varied, relating to fasted and fed conditions [33,53]. At the age of 6–8 months, the gastric emptying is much slower than that in adults, owing to the immature development of neural control for GI motility [54]. Moreover, in the elderly, there is the delayed gastric emptying and a decrease in GI motility [19]. Indeed, women have slower gastric emptying than men, which gives rise to a longer gastric retention time and attenuates drug absorption [34,55,56]. Therefore, in women, a longer interval between having meals and taking drugs is required [56].

Chitosan-poly(acrylic) acid (PAA) polyionic complex was fabricated and was shown to prolong gastric retention of ampicillin, an antibiotic used for the treatment of *Helicobacter pylori* infection-induced gastric ulcer, in swelling and drug release studies, suggesting a role in drug absorption enhancement of this complex [57]. Furthermore, chitosan-based nanoparticles using sodium tripolyphosphate (TPP) as a cross-linking agent were revealed to delay gastric emptying, as well as to improve the absorption and oral bioavailability of ketoconazole, an antifungal drug with poor absorption due to rapid gastric emptying and a short gastric residence time [58]. In addition, in ex vivo porcine gastric mucosa, ketoconazole-loaded chitosan/TPP was demonstrated to adhere to negatively charged mucin layers, implicating good mucoadhesive properties of this nanoparticle formulation [58].

### 2.3. Transport Proteins and Enzymes

The drug substances reaching to the small intestine, where the major site for drug absorption is taken, can be transported across the enterocytes by several mechanisms, such as simple diffusion, active transport, facilitated transport, and pinocytosis, via the paracellular or transcellular route. However, the drugs can be pumped out of enterocytes by the ATP-binding cassette (ABC) efflux transporters, which are a major hindrance for the drug assimilation. The drug transporter proteins located at the apical side of the intestinal epithelial cells include P-glycoprotein (P-gp) multidrug resistance protein-2 (MRP-2) and breast cancer resistance protein (BCRP) [59,60,61,62,63]. These transporters are responsible for the drug efflux into the intestinal lumen. In addition, they can bind to some unspecific compounds other than drug substrates; therefore, they may attenuate the absorption of some drugs, such as antibiotics, lipid-lowering agents, and anti-cancer drugs. Meanwhile, the other transporters located at basolateral side of enterocytes comprise MRP-1, -3, and -5, which are for pumping the drug into the blood circulation [63,64,65,66,67,68]. In the small bowel, the drugs also encounter bile salts and pancreatic enzymes secreted into the intestinal lumen. These bile salts and pancreatic enzymes contribute to the drug dissolution and solubility in the GI tract [69,70]. The intestine-absorbed drugs will pass into the liver via the portal vein in order to undergo drug metabolism. The drug-metabolizing enzymes (DMEs), including cytochrome P450 (CYP450) and CYP3A, also modulate the drug absorption and oral bioavailability [71,72,73]. The immaturity of the GI function in pediatric patients causes the increase in epithelial permeability [33,74]. The intestinal drug absorption in newborns might be variable due to irregular peristalsis. The expressions and functions of efflux transporter protein (i.e., P-gp) and tight junction may increase with age, as the early infancy relies on the passive diffusion [75,76]. However, Toornvliet and colleagues revealed the decreased activity of P-gp efflux transporter in the elderly [77]. For ethnicity-based drug assimilation, the genetic polymorphisms are variedly distributed to a gene product related to drug metabolism enzymes, including CYP450 superfamilies, and transport proteins, including P-gp and organic anion transporting polypeptides [OATPs] [17,78,79]. Several studies have suggested that the P-gp expression is abundantly found in Africans, compared with Caucasians [80,81,82,83].

Interestingly, chitosan can attenuate the P-gp expression in dose-dependent manner, and thereby enhance the oral bioavailability of norfloxacin, an antibacterial agent used for the treatment of gram-negative bacteria infection, in Grass Carp [84]. For the chitosan conjugates, curcumin-carboxymethyl chitosan was developed, serving as an inhibitor of P-gp and an enhancer of drug absorption [85]. In addition, quercetin-chitosan (QT-CS) conjugate was synthesized as an enhancer of drug absorption of doxorubicin by inhibiting P-gp, opening tight junction, and enhancing the water solubility [86]. Carboxymethyl chitosan-quercetin conjugate also helped to improve oral drug delivery and oral bioavailability of paclitaxel by inhibiting P-gp and increasing its water solubility [87].

### 2.4. Gut Microbiota

The human gut microbiota, containing >10^14^ microbes and their genome, resides in the distal segment of the GI tract, particularly in the ileum and colon [88]. For the small intestine, there is less gut microbiota due to its lower luminal pH, higher levels of oxygen delivery, and higher concentrations of antimicrobial agents [89,90,91,92]. The gut microbiota helps to regulate the immune system and to maintain physiological conditions. Importantly, it can ferment the indigested carbohydrates and proteins, giving rise to the byproducts as short-chain fatty acids (SCFAs) [93,94]. Moreover, the gut microbiota and microbial metabolites can metabolize endogenous substances such as amino acids, cholesterol, and bile acids, by various enzymes (e.g., glucuronidases and glucosidases), and modulate the pharmacokinetics (i.e., drug absorption and drug metabolism) [95]. The study of the drugs affected by gut microbiota sheds the light on personalized medicine to predict the drug pharmacokinetics for individual patients. The human microbiome can alter drug-induced pharmacological and toxicological effects. For example, the gut microbiota can alter the oral bioavailability and activity of insulin, as it is susceptible to proteolysis [92,96]. Additionally, it may increase acetaminophen-induced hepatotoxicity, involved with p-cresol, a microbial metabolite [97]. The enhanced toxicity of diclofenac by gut microbiota is associated with the deglucuronidation and delayed excretion [98]. The decrease in firs pass a metabolism effect and high levels of intestinal β-glucuronidase activity were documented in neonates [99]. The gender difference contributing to the gut microbiota compositions has not been well-addressed due to its inconsistency. However, several lines of evidence related to the sex difference in gut microbiota have been documented [100]. Ethnicity-, dietary-, and lifestyle-specific variations in gut microbiota composition are also observed [101,102].

Several studies have reported that chitosan and its derivatives can modulate the gut microbiota imbalance. For instance, carboxymethyl chitosan has revealed to alter the gut microbiota in *Escherichia coli* (*E. coli*)-treated mice, affecting to fat and glucose metabolism, as well as the inflammatory profile [103,104]. It was also found that chitosan-chelated zinc has reduced the intestinal inflammatory process and mucosal injury in *E. coli*-infected rats by reversing the gut bacterial composition [105]. In addition, chitosan oligosaccharide (COS), a derivative of chitosan, has been exhibited to restore the gut microbial imbalance in diabetic mice [106].

### 2.5. Disease Conditions

The disease conditions can influence the drug absorption of orally administered drugs, as they impact on the structures and functions of GI tract organs, including esophagus, stomach, small intestine, and large intestine. The change in drug absorption is associated with the alteration of the aforementioned physiological characteristics.

In diabetic patients, they secrete less gastric acid than healthy individuals [107]. Therefore, the gastric pH is increased, resulting in the poor absorption of basic drugs. However, the delayed gastric emptying and prolonged transit times have been impressed in diabetes [108,109,110]. Importantly, owing to the microvascular complications, the gastric blood flow is declined, leading to further delayed gastric emptying and affect to the drug absorption in the small bowel. The alterations of P-gp expression and function under diabetic conditions were decreased in diabetic rats [111]. However, some evidence has mentioned that the P-gp expression is temporally induced and then restored to a baseline level [112]. In addition, it has been shown that the level of the CYP3A4 enzyme is reduced, leading to the change in the oral bioavailability of some drugs related to CYP3A4 activity, such as carbamazepine, anti-retroviral drugs and some statins [113,114]. In obesity, a larger gastric volumes and lower gastric pH have been reported, compared with lean subjects [115]. The gastric emptying time in obese patients is controversial [25]. For the intestinal transporter proteins, the inhibition of P-gp could result in hepatic steatosis and obesity using P-gp deficiency mice fed a high-fat diet model [116]. The CYP34A enzyme activity was found to be reduced in obese individuals; therefore, it needs to be cautious when using CYP3A4 substrates, inhibitors, or inducers in these patients [117]. For the treatment of diabetic patients, oral insulin is applied due to its greater convenience and its less adverse drug reaction than subcutaneously injected insulin. However, it still interferes with proteolytic degradation and mucosal layers in the GI tract. Insulin-loaded chitosan nanoparticles increase residence time to retain in the GI tract and enhance mucoadhesion, therefore promoting drug assimilation [118,119].

In patients with gastric cancer, the gastric pH is 6–7, which results from the gastric atrophy and decreased gastric secretion [120,121]. The gastric emptying is slower than in healthy subjects, and is further decreased after gastrectomy [122,123]. However, the oral anticancer drugs may improve their efficacy as they retain in the stomach for a longer duration, thus providing sufficient time for gastric tumor to expose with drugs [124]. For colorectal cancer, the transit time is not clear, although the colon motility change in this type of cancer is well-known [25]. Of note, the efflux transporters (e.g., P-gp, MRP-2, BCRP) were reported to be altered in cancer cells and were still the big concern for the drug resistance in the cancers [125,126]. Doxorubicin is an oral chemotherapy used in the treatment of considerable cancers, including lung, gastric, breast, thyroid, and ovarian cancers, by acting on the nucleus of target cells [127,128,129]. Doxorubicin was documented to have a low bioavailability because it was abolished by the first-pass metabolism of CYP450 and the overexpression of P-gp [130,131,132,133]. It was shown that doxorubicin-loaded chitosan nanoparticles increased the permeation across intestinal epithelium. Moreover, when conjugating with quercetin, it inhibited the P-gp efflux transporter. Collectively, chitosan was claimed to relieve the poor absorption of doxorubicin [87,131].

The effect of GI tract infections on oral drug assimilation is unpredictable, as the GI membrane damage induced by infections can cause both the increase and decrease in drug intestinal absorption. The delayed gastric emptying may be associated with an increasing chance of experiencing nausea and vomiting and a longer onset of medications, resulting in uncontrolled drug plasma concentrations. *H. pylori* infection leads to the decrease in gastric secretion and the impairment of drug absorption. The poor absorption of some antibacterial drugs (e.g., gentamicin, metronidazole) were alleviated by the drug-chitosan complex [134,135]. For pain and stress, the alterations of GI physiology are relevant to the gut–brain axis, such as the reduction in GI motility, gastric secretion, and mucosal blood being low [136]. It was demonstrated that chitosan-based hydrogels could manipulate the drug release of paracetamol in the intestine [137]. Besides, a chitosan complex could regulate the drug release properties of ibuprofen, a non-steroidal anti-inflammatory drug (NSAID) [138].

The altered kinetics of drug absorption in these disease states have been partially acknowledged. Further investigations involving pharmacokinetics towards personalized and precision medicines should be considered.

## 3. Chitosan-Based Nanomaterials for Improving Intestinal Drug Absorption and Their Pharmacological Applications

Since one of our goals of this review is to summarize the types of fabricated chitosan-based nanomaterials in the aspect of the drug absorption enhancer, the literature search was performed on the full-text articles published in English, with date limits of 2010 up to 8 February 2021, assessed in the PubMed database by using the search term of “chitosan” AND “intestine” AND “drug delivery”. As a result, a total of 106 articles were collected by EndNote and further selected for relevant topics. Additional articles that provided specific evidence and information (e.g., diabetes, cancers, infections, inflammation, polyelectrolyte complex, thiolated chitosan, trimethyl chitosan, carboxymethyl chitosan, alginate, Eudragit etc.) were included.

Chitosan-based nanomaterials have received tremendous attention from many investigators over the past decade as the enhancer of drug delivery, particularly drug absorption, based on their marvelous properties, which are (i) protection against GI luminal degradation (ii) mucoadhesion, (iii) permeation enhancement, (iv) controlled drug release, and (v) efflux inhibition (Figure 2). The significance of chitosan-based nanomaterials is indicated and discussed in detail as below. In addition, the types of chitosan-based nanomaterials are summarized in Table 1.

### 3.1. Chitosan-Based Polymeric Nanoparticles with Chemical Modifications

Although chitosan is well-soluble in acidic medium, it has poor solubility in the basic environment, as in the duodenum. Fortunately, chitosan is easy to chemically modify at its functional groups, including an amino group (-NH_2_) and two hydroxyl groups (-OH), in order to improve its physical and biological properties [139].

Thiolated chitosan is a favorable derivative of chitosan with the immobilized sulfhydryl- or thiol-bearing groups onto the primary amine groups of chitosan. The disulfide bond formation with cysteine-rich subdomains of mucus glycoprotein by a thiol-disulfide interchange reaction leads to the close contact between the thiomer and mucosal layers, hence making this modified chitosan have a better mucus entrapment efficiency, compared with conventional chitosan [139]. The increased mucoadhesivity retains the nanoparticles at the site of absorption, giving rise to the enhanced drug oral bioavailability. Fabiano and colleagues revealed the importance of thiol groups on drug oral bioavailability by comparing quaternary ammonium-chitosan with S-protected thiol groups (NP QA-Ch-S-pro) and without thiol ones (NP QA-Ch) [140]. Moreover, thiolated chitosan provides a beneficial effect as a permeation enhancer by inhibiting the protein tyrosine phosphatase (PTP) enzyme, which in turn dephosphorylates the tyrosine subunits of occluding protein and further promotes the opening of tight junctions [141,142]. The enhancement of permeation depends on the degree of the thiolation and the physiochemical properties of the drug [143]. The utilizations of thiolated chitosan were denoted in diabetes treatment. In the insulin delivery, oral insulin-loaded thiolated chitosan nanoparticles (Ins-TCNPs) could be prepared with pentaerythritol tetrakis (3-mercaptoproprionate) or PETMP, which was used for penetrating the cell membrane, while the thiomer inhibited the PTP enzyme and enhanced the opening tight junctions, promoting the drug absorption [118]. In addition, Ins-TCNPs was shown to reduce blood glucose as well as enhance the level of plasma insulin and prolong its duration in rats [118,144]. Insulin was released from thiolated chitosan-based nanoparticles at two phases, which were a rapid initial release or burst release at pH 2 and a sustained release at pH 5.3 [145,146]. It also improved the mucus adhesion between the polymer and the intestinal mucosa and prolonged drug gastric residence time, improving the oral bioavailability and thus implicating the potential role of thiolated chitosan in oral delivery of insulin [118]. Besides mucoadhesive and permeation-enhancing features, thiomers could inhibit the ATP-dependent drug efflux pumps, including P-gp and MRP-2, which were able to pump the drug molecules out of the cells, causing the attenuated drug efficiency and the poor oral drug bioavailability. Recently, it has been shown that both anionic and cationic thiolated chitosans could inhibit these efflux pumps and promote intestinal transcellular drug uptake [147]. For their applications in cancers, docetaxel (DTX), an anticancer drug for metastatic breast, lung, and gastric cancers, was carried by thiolated chitosan with the improvement of cellular internalization, the augmentation of mucoadhesivity, and the inhibition of P-gp [148]. Aside from DTX, α-mangostin, a natural xanthonoid extracted from mangosteen, was loaded by thiolated chitosan with pH-dependent Eudragit L100 and cross-linking genipin and this xanthonoid-bearing nanoparticle exhibited the mucoadhesive and controlled drug release properties in colorectal cancer [149]. Besides, thiolated chitosan could provide to increase the stability of low-molecular weight heparin (LMWH) in a gastric environment by ionic cross-linking with hydroxypropyl methylcellulose phthalate (HPMCP) [150]. Mechanistically, thiolated chitosan improved mucoadhesion, increased paracellular permeability of LMWH at the absorption site, and controlled the release of LMWH in the circulation [150]. Moreover, S-protected chitosan-thioglycolic acid using 6-mercaptonicotinamide (TGA-MNA) had better mucoadhesive, permeation-enhancing, and efflux pump-inhibiting properties than chitosan-thioglycolic acid (chitosan-TGA) with thiol-free groups [151,152,153]. Thiolated chitosan with reduced glutathione (GSH) enhanced the oral drug delivery of leuprolide, a poor-permeable peptide drug using as a gonadotropin-releasing hormone (GnRH) analogue, by preventing it from enzyme degradation and promoting its absorption [154,155].

Trimethyl chitosan (TMC) is a water-soluble quaternized chitosan derivative produced from the methylation of chitosan with iodomethane in the presence of sodium hydroxide (NaOH) at the elevated temperature [156,157]. TMC has the potential for augmenting the membrane-penetrating features for hydrophilic macromolecular drugs, which are poorly absorbed owing to rapid hydrolytic degradation by gastrointestinal juices [158]. For the treatment of diabetes, encapsulated insulin-incorporated TMCs exerted mucoadhesive and permeation-enhancing effects and stimulated the intestinal absorption by mediating the intestinal barrier integrity and promoted the insulin transport via the paracellular route. The self-assembly of chitosan nanoparticles with negatively charged fucoidan was anticipated to have the hypoglycemic effects and avoid diabetic complications [159]. Moreover, it was documented that oral insulin-loaded TMC nanoparticles provided a better therapeutic outcome, compared with injected insulin in type 1 diabetes (T1D) [160]. In the field of cancers, TMC could be applied in the drug delivery of paclitaxel (PTX), an anti-microtubule agent using the treatment of gastrointestinal tumors. TMC-loaded PTX enhanced the internalization of the nanoparticles and exerted the cytotoxic effect to cancer cells without the systemic adverse effects [161]. Interestingly, when TMC-loaded peptides were coated with liposome, they prolonged the residence time and increased the mucoadhesivity in the GI tract [162]. TMCs were also utilized in peptide drug delivery in cancer immunotherapy. For instance, the Oral PD-L1 Binding Peptide 1 (OPBP-1)-loaded TMC was shown to increase the oral bioavailability of the peptide drug and impede the tumor cell growth [163]. In addition, it has been reported that TMCs could carry curcumin, which plays a vital role in the signaling cascades related to cancers and inflammation, as curcumin-loaded TMC provided the controlled release and enhanced the oral bioavailability of curcumin [164,165]. Interestingly, TMC-coated solid lipid nanoparticles (SLNs) incorporated the curcumin displayed in the sustained release of curcumin and elevated its oral bioavailability, compared with free curcumin and non-coating chitosan nanoparticles [166].

Carboxymethyl chitosan (CMS or CMCS) is a pH-sensitive chitosan derivative with the carboxymethyl substituent that improves the water solubility in a neutral and basic pH, based on the degree of substitution, as well as enhancing the mucoadhesive properties [167]. CMCS can be produced by direct or reductive alkylation. For the treatment of diabetes, insulin-loaded chitosan/CMCS nanogels increased mucoadhesion and permeation in a rat jejunal ex vivo model, even though it had no differences in the duodenum [168]. CMCS has a negative charge, which can interact with positively charged chitosan to obtain the polyelectrolyte complex through electrostatic interaction, which provides the pH responsive stability and thus controls the drug release [133]. This will be included in the later section “chitosan-based polyelectrolyte complex nanoparticles”. Several lines of evidence have shown that doxorubicin, an anti-neoplastic drug used for treating solid tumors, can be loaded with chitosan/*O*-carboxymethyl chitosan (OCMCS) by simple ionic gelation. Doxorubicin-loaded chitosan/CMCS nanoparticles had better mucoadhesive properties and helped to improve oral bioavailability in ex vivo intestine [133,169,170]. Furthermore, CMCS was employed in the delivery of clarithromycin for the treatment of *H. pylori*. infection in the gastric environment. CMCS was grafted with stearic acid and conjugated with urea to obtain ureido-modified CMCS-graft-stearic acid (U-CMCS-g-SA). This nanopolymer was claimed to have a gastric retention property and a drug-controlled release with a target specificity [171]. For the inflammation, pH-responsive OCMCS/fucoidan nanoparticles were potentially used for the delivery of curcumin, as they controlled the release of curcumin and promoted the cellular uptake and endocytosis of curcumin Caco-2 cells [165]. In addition, gum Arabic (GA)-CMCS microcapsules enhanced the oral bioavailability of omeprazole, an antacid and a proton-pump inhibitor, to augment GI-targeted delivery [172].

### 3.2. Conjugated Chitosan

Chitosan can conjugate or form the complex with other superporous networks and (SPNs)-based polymers, which have numerous interconnected pores in three-dimensional structures with water-soluble agents, such as poly(vinyl alcohol) (PVA) and alginate [173]. The rapid swelling of SPNs could occur since the porous space of SPNs provide the water absorption, extending the gastric retention of the drug in the GI tract [174]. For the applications of conjugated chitosan, cross-linked chitosan/PVA was revealed to enhance the delivery of ascorbic acid with the formulated hydroxypropyl methylcellulose (HPMC), a hydrophilic and polymer and glyoxal, which was used for a crosslinking reagent, by prolonging the gastric retention time and promoting sustained release [173]. Additionally, for the diabetes therapy, the conjugation of cationic charged chitosan and anionic charged poly(γ-glutamic acid) (PGA) ameliorated the oral insulin delivery to adhere to the GI mucosal surface and transiently stimulated to open the tight junctions of the intestinal epithelial cells [175,176]. Moreover, in the treatment of cancer, Caco-2 cell monolayers treated with antiangiogenic, protein-loaded chitosan-*N*-arginine/PGA-taurine conjugated nanoparticles exhibited the enhancement of the cell permeation of antiangiogenic protein [177]. Not only chitosan alone, but chemically modified chitosans could take part in the conjugation process. For example, CMCS was prepared with poly(ethylene glycol) (PEG) to load doxorubicin hydrochloride in tumor cells. This nanoparticle was able to serve as a candidate for drug delivery of the antitumor drug [178]. Besides, thiolated chitosan could be conjugated with PEG to generate the thiol group bearing PEGylated chitosan (Chito-PEG-SH) with magnified mucoadhesive and permeation-enhancing properties, in isolated porcine and rat intestines [179]. In addition to drug delivery, chitosan-based nanoparticles are employed in a protein delivery system by inventing spherical PEG-grafted (chitosan-g-PEG), using TPP or PGA as a cross-linking agent [180].

### 3.3. Chitosan-Based Polyelectrolyte Complex Nanoparticles/Nanocapsules

To improve the drug stability in the GI tract and to lessen the systemic toxicity, the nanoencapsulation of the drugs is generated as the drug molecule loading into nanocarriers (e.g., nanoparticles, liposomes, micelles, and microemulsions) [130,132,181,182,183,184]. However, the encapsulated drugs confront the site-dependent pH change along the GI tract as the pH increases from the stomach (pH 1.2–3.7) to the small intestine (pH 6–7.4). Therefore, pH-sensitive/responsive polymers containing ionizable acidic and basic residues regarding the environmental pH are produced. The formations of the polyelectrolytes by non-covalent electrostatic interactions between polycations and polyanions are known as polyelectrolyte complex nanoparticles (PECNs). In response to the pH variation, these complex polymers could be adapted as swelling or shrinking, relying on the different degree of ionization of functional groups of PECNs [185]. The multishell structures in hollow nanocapsules were made from the dissolution of the core from the epitaxial core-shell structure of the nanoparticles [185,186]. A polyelectrolyte sphere was developed by the layer-by-layer (LbL) assembly method, employing a positively charged chitosan as the pH-responsive outer layer. The shrinking process of the chitosan-based nanoparticle might occur in the acidic surroundings [187,188].

Apart from chitosan, other biocompatible polymers, such as alginate, tripolyphosphate (TPP), and Eudragit, can serve as the outer membrane of nanocapsules. Alginate, consisting of 1,4-linked β-D-mannuronic acid and α-L-guluronic acid, was also a water-soluble and biodegradable natural polymer derived from seaweed [137,189]. It was able to crosslink with divalent/polyvalent cations (e.g., calcium, zinc, and copper) to form a network structure for use in sustaining drug release [137,189]. The carboxyl groups of alginate provided the negative charge and interact with the positively charged amino group of chitosan in the polyelectrolyte complex gel. Methoxy poly(ethylene glycol) or poly(ethylene glycol) monomethyl ether (mPEG)-grafted-CMCS (mPEG-g-CMCS) was synthesized with alginate to increase the loading capacity and provide a well-controlled drug release, particularly in the basic environment. In other words, mPEG-g-CMCS was a promising pH-sensitive nanoparticle for site-specific drug delivery in the intestine [190]. It was demonstrated that the fabricated core-shell nanoparticles, containing the curcumin at the core and the chitosan/alginate multilayer shell, could exhibit controlled release of the curcumin nanocrystal to target the inflamed colon in mice with ulcerative colitis (UC) [191]. In addition to alginate, TPP has been well-recognized. It had a negative charge and was used as a cross-linking agent with chitosan for the application of a chitosan-based hybrid system, such as quercetin-loaded and 5-Flurouracil (5-FU)-encapsulated chitosan/TPP nanoparticles [192,193,194]. Apart from that, a Eudragit polymer could be used for coating the nanoparticles in order to prevent the rapid release of curcumin and theophylline in the stomach and small intestine [195,196].

Chitosan-based PECNs are prepared from the interaction between the opposite charges of copolymers. For instance, chitosan/insulin PECNs could be obtained by positively charged chitosan-g-mPEG copolymers and negatively charged insulin [197]. These PECNs were used for improving the oral insulin to cross the mucus and epithelial membrane barriers in the GI tract [197,198]. mPEGylation is the process to enhance mucus-penetrating properties of chitosan-based nanomaterials. Recently, mPEG_10%_-chitosan glyceryl monocaprylate (GMC)_10%_ copolymers have been modified to add the feature of hydrophobicity onto their surface, which helped to impale the epithelial membrane [197]. However, the differential advantages of hydrophilicity/hydrophobicity were still controversial for preferable usage of nanoparticles, since the hydrophilic surface was better permeable in the mucus layer, while the hydrophobic exterior was preferred to penetrate the cell membrane.

In addition, pH-sensitive PECNs were generated for improving doxorubicin to overcome the obstacles both in the GI tract and the intracellular tumor cell regulation, such as multidrug resistance (MDR) [199]. Chitosan/doxorubicin PECNs contained two polyelectrolytes, including positively charged chitosan and negatively charged poly(L-glutamic acid) grafted polyethylene glycol-doxorubicin conjugate nanoparticles (PG-g-PEG-hyd-DOX NPs) [132]. These PECNs prolonged duration in the blood circulation and tumor tissue accumulation with rapid drug release at target cells; therefore, these PECNs might serve as the ideal nanocarriers for the anticancer field [132]. Additionally, chitosan-based PECNs could be synthesized by the electrostatic interactions between positively charged chitosan and negatively charged CMCS, which was a water soluble derivative of chitosan, as previously mentioned, and produce the nanoparticles by the ionic gelation using CaCl_2_ as a cross-linking agent [131]. Doxorubicin was also prepared as DOX-loaded chitosan/carboxymethyl chitosan-based nanoparticles (DOX:CS/CMCS-NPs) with the porous-core nanoparticle and coacervate microcapsules-immobilized multilayer sodium alginate beads (NPs-M-ALG-Beads and CMs-M-ALG-Beads) in order to stabilize the drug in the GI tract [133,169,170].

Furthermore, the delivery of several antimicrobial agents was enhanced by the PECNs. For example, delafloxacin, a broad-spectrum fluoroquinolone used for both Gram-positive and negative aerobic and anaerobic bacteria, was loaded with positively charged chitosan and negatively charged stearic acid. The hybrid nanoparticles could increase the bioavailability and sustain the drug release via the electrostatic interaction of the polymer and lipid [200]. Another fluoroquinolone ciprofloxacin was carried by an embelin-chitosan gold nanoparticle (Emb-Chi-Au). This complex nanoparticle could enhance antipseudomonal activities by inhibiting the MDR efflux pump [201]. Tobramycin, an aminoglycoside antibiotic used for inhibiting biofilm formation by *Pseudomonas aeruginosa*, also relied on PECNs using *N*,*O*-[*N*,*N*-diethylaminomethyl(diethyldimethylene ammonium)*_n_* methyl]chitosan (QAL) nanoparticles and GENUVISCO type CSW-2 carrageenin (CG) [202]. Moreover, to achieve a better anti-inflammatory effect, curcumin was loaded with chitosan/GA and it was found that this nanoparticle helped control the release of curcumin [165]. Another example of CMCS-based application was the drug delivery of omeprazole using gum Arabic (GA)-*O*-carboxymethyl chitosan LbL microcapsules, which were able to enhance the oral bioavailability of omeprazole [172].

**Table 1 pharmaceutics-13-00887-t001:** Summary of chitosan-based nanocarriers used for improving the absorption of indicated drugs and their pharmacologic effects. This table summarizes the chitosan-based nanomaterials that carry the considerable drugs for better drug assimilation. The nanocarriers, loading drugs, and their exerting pharmacological effects are listed with their attainable references. Abbreviations: LMWH, low-molecular weight heparin; OPBP-1, Oral PD-L1 Binding Peptide 1; 5-FU, 5-flurouracil; BSA, bovine serum albumin.

Nanocarrier	Drug	Pharmacological Effect(s)	Reference(s)
1. Chemical modification			
1.1 Thiolated chitosan	Insulin	mucoadhesion, permeation enhancement, controlled drug release	[118]
	Docetaxel	mucoadhesion, permeation enhancement, controlled drug release, efflux inhibition	[148]
	α-mangostin	mucoadhesion, controlled drug release	[149]
	LMWH	protection against GI luminal degradation, mucoadhesion, permeation enhancement, controlled drug release	[150]
	Leuprolide	mucoadhesion, permeation enhancement	[154]
1.2 Trimethyl chitosan (TMC)	Insulin	mucoadhesion, permeation enhancement, controlled drug release	[160]
	Paclitaxel	mucoadhesion, permeation enhancement, controlled drug release	[161]
	Calcitonin	mucoadhesion, permeation enhancement, prolongation of residence time	[162]
	OPBP-1	mucoadhesion, permeation enhancement, controlled drug release	[163]
	Curcumin	mucoadhesion, permeation enhancement, controlled drug release	[164,165,166]
1.3 Carboxymethyl chitosan	Doxorubicin	mucoadhesion, permeation enhancement, controlled drug release, efflux inhibition	[131,132,133,169,170]
	Clarithromycin	controlled drug release, prolongation of residence time	[171]
	5-FU	controlled drug release	[192]
	Curcumin	mucoadhesion, controlled drug release, efflux inhibition	[85,165]
	Omeprazole	protection against gastric degradation, controlled drug release	[172]
2. Conjugation			
Poly(vinyl alcohol) (PVA)	Ascorbic acid	controlled drug release	[173]
Poly(γ-glutamic acid) (PGA	Insulin	protection against GI luminal degradation, mucoadhsion, permeation enhancement	[175,176]
Poly(ethylene glycol) (PEG)	Insulin	mucoadhsion, permeation enhancement, controlled drug release	[197]
	BSA	mucoadhsion, permeation enhancement, controlled drug release	[190]
3. Polyelectrolyte complexation	Insulin	protection against GI luminal degradation, mucoadhesion, permeation enhancement, controlled drug release	[119,197]
	Doxorubicin	mucoadhesion, permeation enhancement, controlled drug release, efflux inhibition	[131,132,133,169,170]
	5-FU	controlled drug release	[192]
	Quercetin	protection against GI luminal degradation, controlled drug release	[193]
	Curcumin	mucoadhesion, controlled drug release	[165,191,195]
	Rutin	mucoadhesion, permeation enhancement, controlled drug release	[194]
	Gentamicin	mucoadhesion, permeation enhancement, controlled drug release	[134]
	Paracetamol	permeation enhancement, controlled drug release	[137]
	Ibuprofen	controlled drug release	[138]
	Omeprazole	protection against GI luminal degradation, controlled drug release	[172]
	Furosemide	mucoadhesion, permeation enhancement, controlled drug release	[189]
	Theophylline	controlled drug release	[196]
	Delafloxacin	controlled drug release	[200]
	Ciprofloxacin	efflux inhibition	[201]
	Tobramycin	mucoadhesion, permeation enhancement	[202]

## 4. Future Perspectives and Conclusions

The oral route is the most promising for drug administration; however, it is essential to overcome various physiological barriers along the GI tract (i.e., gastric pH, GI enzymes, mucus layer, efflux pump) before reaching the systemic circulation and exerting its effects at the action site. These challenges limit the absorption of multitudinous drugs, including antidiabetic, anticancer, antimicrobial, and anti-inflammatory drugs and hence their oral drug bioavailability. To surpass these GI obstacles, the drug delivery carriers have been designed and developed. Chitosan is chosen to be a potentially unique candidate that meets the ideal properties for bioinspired drug delivery. Chitosan is a natural origin-based polymer with its versatile functional groups, and hence it is feasible to be chemically modified to improve the chitosan’s physiochemical properties, including mucoadhesive capabilities, the permeating-enhancing effect, controlled drug release, and efflux inhibition. Several types of chitosan-based nanomaterials are fabricated and they are investigated for the potential to be the enhancer for drug assimilation. Even though the drug can reach the blood circulation, the fluctuations in drug concentrations can occur and may result in the increased incidence of either the drug adverse reaction or subtherapeutic treatment. Therefore, the sustained delivery system has been introduced to solve this problem [9,203]. Thiolated chitosan can be applied for the sustained delivery, but the maintenance of the drug remaining for longer period of time is required. Moreover, particularly in the elderly, polypharmacy is increasingly occupied in many cases. For the effective treatment, the co-delivery systems will assist to carry multiple drugs to differential targets at the same time [204,205,206]. Aging-, gender-, and race-dependent physiological factors related to drug absorption are still unclear. A better understanding in the study of personalized medicine will help to clarify these variations. The state-of-art technology related to personalized medicine is needed to be developed. Considering the novelty of theranostics, the combination of the therapeutics and diagnostics, chitosan and chitosan-based nanocarriers have been emerged in the biomedical engineering field. The large varieties of chitosan derivatives and modifications, as well as their versatilities, allow them to be the ideal carriers that can reach the specific site of such diseases. Theranostics and imaging-guided therapies will help instantaneously keep track of chitosan-based drug delivery and direct them to the target site [207]. The development of personalized nanomedicine will provide the information of an individualized drug response in order to adjust it to an optimal dosage and proper management for each patient [208]. Moreover, since chitosan could be applied with other polymers to fabricate various chitosan formulations, the studies of toxicity and safety of chitosan-based nanoparticles need to be further explored in vivo to ascertain the appropriate dose selection in humans [209]. The in-depth in vivo studies for chitosan nanoparticles may include the types of utilized polymer, size, shape, surface, morphology, and electrokinetic potential, which are significant determinants for the toxicity and safety and are associated with the nanotoxicity of these nanoparticles [207]. It is possible that the cytotoxicity of chitosan-based nanocarriers is resulted from the electrostatic interaction between chitosan and cell membrane, as well as the cellular uptake of chitosan and subsequent activation of intracellular signaling cascades [210]. Unfortunately, for the biocompatibility studies in the animal models, the short duration after the intravenous injection of nanoparticles might be insufficient [207]. Therefore, the clinical efficiency and in vivo efficacy of these nanoparticles should be stepped up. Another concern in the use of chitosan-based nanomaterials is that the antimicrobial activity of chitosan may be disrupted by interaction with food (e.g., fruits and vegetables). Besides, the patients who take warfarin, an anticoagulant, with chitosan, may have a potential risk of bleeding due to the effect of chitosan on intrinsic coagulation and fat-soluble vitamin absorption [211]. In addition, for the pharmaceutically industrial and commercial scales, the technology for the high-throughput level may be integrated. Recent advances in the development and application of chitosan-based nanomaterials have been brought up in this review. The implementations of the nanomaterials in diabetes, cancers, infections, and inflammation are specific; therefore, the further research for polymeric drug delivery in such individual diseases are encouraged toward a new paradigm in the future treatment based on nanotechnology.

## Figures and Tables

**Figure 1 pharmaceutics-13-00887-f001:**
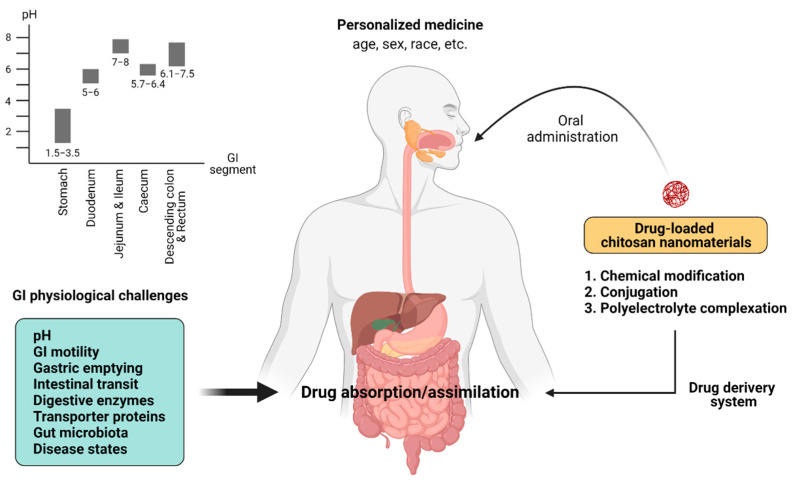
The schematic illustration of the GI physiological barriers affecting drug absorption. When one orally takes the drug-loaded chitosan nanomaterials (e.g., chitosan-based nanoparticles with chemical modification, conjugated chitosan, and chitosan-based polyelectrolyte complex), they encounter GI physiological challenges, including a variable pH along the GI tract, GI motility and gastric emptying, intestinal transit, digestive enzymes, transporter protein expression, gut microbiota, and disease conditions. The GI physiological challenge can be influenced by aging, gender, and ethnicity. Created with BioRender.com.

**Figure 2 pharmaceutics-13-00887-f002:**
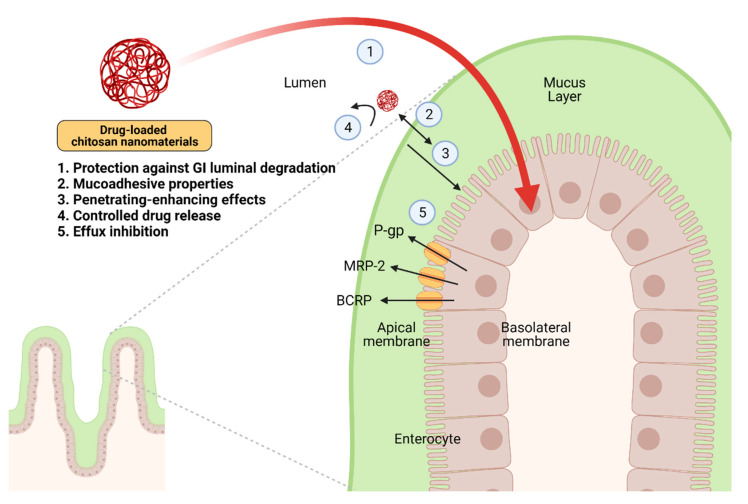
The mechanistic insights of the enhancement of drug absorption by drug-loaded chitosan nanomaterials in the intestine. The ideal properties of these nanomaterials include (i) protection against GI luminal degradation, (ii) mucoadhesion, (iii) permeation enhancement, (iv) controlled drug release, and (v) inhibition of P-gp, MRP-2, or BCRP. Abbreviation: P-gp, P-glycoprotein; MRP-2, multidrug resistance protein-2; BCRP, breast cancer resistance protein. Created with BioRender.com.

## Data Availability

Not applicable.

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
