# Peer review of "Potential Applications of Chitosan-Based Nanomaterials to Surpass the Gastrointestinal Physiological Obstacles and Enhance the Intestinal Drug Absorption"

_pharmaceutics, 2021, doi:10.3390/pharmaceutics13060887_

Round 1
Reviewer 1 Report
Well organized and structured review.
Small Comments.
- Is important to give the perspective and the challenges of the future on the field of chitosan based materials as per-os drug delivery system.
- Many examples are out there on the evolution of nanopharmaceutical formulations of Chitosan as materials therefore and potential application on ph response drug delivery systems not only for small chemical structure drugs though also for proteins e.g Papadimitriou et al. International Journal of Pharmaceutics 430, (1-2), 318-327.
Reviewer 2 Report
The manuscript is well written and presented. However, I suggest the following changes
- Please include some recent results. Only 10 papers in 2020?
- Revise section three with "diabetes, cancers, infections, and inflammation" centered narrative.
Reviewer 3 Report
The manuscript entitled: "Chitosan-based Nanomaterials as Enhancers of Intestinal Drug Absorption" present information about the physiological challenges affecting intestinal drug absorption and the effects of chitosan on those parameters impacting on oral bioavailability.
Since the chitosan-based nanomaterials are not in the main attention of the authors, I believe that should change the manuscript title, or otherwise to add a section discussing about the chitosan-based nanomaterials preparation. In this case a schematization of the main procedures to obtain these nanomaterials.
section 2.1. Gastric pH is generally well written, but more information needs to be included about the challenges of pH.
Line 218: "modificaitons" please correct
Round 2
Reviewer 2 Report
The authors have addressed the reviewer's concerns.